# Application of Differential Subsampling with Cartesian Ordering in Evaluating Left Ovarian Venous Reflux for Pretreatment Planning for Pelvic Venous Disorders

**DOI:** 10.3390/diagnostics14161737

**Published:** 2024-08-10

**Authors:** Sheida Ebrahimi, Nawal Siddiqui, Alexandra Besser, Ana E. Rodriguez-Soto, Hon Yu, Christine Boone, Albert Hsiao, Anne C. Roberts, Rupal Parikh, Rebecca Rakow-Penner

**Affiliations:** 1Department of Radiology, University of California San Diego, La Jolla, CA 92093, USA; 2School of Medicine, University of California San Diego, La Jolla, CA 92093, USA; 3Department of Bioengineering, University of California San Diego, La Jolla, CA 92093, USA

**Keywords:** dynamic contrast-enhanced MRI, diagnostic imaging, pelvic venous disorder, pelvic pain

## Abstract

The diagnosis of a common cause of chronic pelvic pain can be made by visualizing reflux in the ovarian veins. Fluoroscopic venography is the gold standard for diagnosing ovarian vein reflux, but it is an invasive technique that exposes patients to ionizing radiation. MRI, with its lack of ionizing radiation and capability of high-temporal and spatial-resolution vascular imaging, has the potential to provide similar diagnostic information. This retrospective report describes and assesses the utility of a dynamic contrast-enhanced MRI technique based on Differential Subsampling with Cartesian Ordering (DISCO)–MRI in 30 patients with chronic pelvic pain. Among the 14 patients who underwent both DISCO–MRI and fluoroscopic venograms, 11 (78.6%) exhibited concordant results, while 3 patients (21.4%) had discordant findings. These results suggest the potential of multiphasic contrast-enhanced DISCO–MRI as a non-invasive diagnostic tool for evaluating chronic pelvic pain.

## 1. Introduction

Pelvic venous disorders demonstrate a range of symptoms, mainly due to both the reflux of the gonadal and internal iliac veins and the obstruction of the left renal and iliac veins. Multiple symptoms and various pathophysiological mechanisms may present concurrently. For instance, primary ovarian vein reflux, compression of the left common iliac vein, or compression of the left renal vein can all contribute to chronic pelvic pain, which is one of the clinical presentations of pelvic venous disorders [1].

Chronic pelvic pain is a common condition among women, with an estimated prevalence between 5.7 and 26.6% [2,3]. Pelvic venous disorders are an often overlooked and difficult-to-diagnose cause of chronic pelvic pain. Symptoms can be characterized by dull, achy pelvic pain, often unilateral or bilateral, associated with activity and time of day, and possibly associated pain with intercourse that persists for more than 6 months without an identifiable cause [3]. May–Thurner, nutcracker, and pelvic congestion syndromes exhibit overlapping symptoms, and it is recommended that these historical terms no longer be used [4].

Since ovarian vein reflux and dilated ovarian and para-uterine veins can be observed in asymptomatic patients, standard imaging, such as ultrasound or computed tomography, alone is insufficient to diagnose pelvic venous disorders [3]. Therefore, clinical correlation and advanced imaging techniques are helpful for accurate diagnosis in this patient group. Ovarian vein diameter is a poor predictor of ovarian vein reflux. Fluoroscopic venography has been used for imaging workups prior to interventional management [5,6]. However, non-invasive imaging plays an important role in characterization. Ovarian vein reflux can be treated with image-guided intervention and is often evaluated with MRI prior to treatment. Identification of pelvic venous disorders allows for procedure planning for minimally invasive embolization ± sclerotherapy in the setting of ovarian vein reflux, stent placement in the setting of left common iliac vein compression, and possibly surgical management in the setting of left renal vein compression [7].

Ovarian vein reflux can be suggested by reflux on time-resolved MRIs or fluoroscopic venograms, but not necessarily both. These imaging findings may be seen incidentally in asymptomatic patients. High-temporal resolution dynamic time-resolved MR angiography (TR-MRA) is a valuable addition for the evaluation of ovarian vein reflux. TR-MRA has been demonstrated as an accurate method for the evaluation of ovarian venous reflux [7,8]. However, it lacks the spatial resolution available with newer MRI techniques. DISCO (Differential Subsampling with Cartesian Ordering) is a dynamic contrast-enhanced MRI technique that allows for both high temporal and spatial resolution [9].

The technique has shown utility in liver imaging, particularly in hepatocellular carcinoma and hypervascular metastasis [9]. In this study, DISCO was used to evaluate ovarian venous reflux. The overarching goal is to provide a recommended MRI protocol that may aid in pre-procedural planning when the radiologist can prospectively see reflux on a non-invasive study prior to interventional embolization.

## 2. Materials and Methods

### 2.1. Patients

A retrospective evaluation was performed on studies from August 2016 to December 2021 with the following criteria: abdominopelvic magnetic resonance imaging in female patients with indications including concern for chronic pelvic pain and pelvic congestion syndrome. It is important to note that ‘pelvic congestion syndrome’ is an outdated term, recently renamed pelvic venous disorders, but is part of our historical database. The DISCO sequence is part of our standard protocol for the evaluation of pelvic venous disorders. A waiver of consent was granted from the Institutional Review Board for this retrospective study. A total of 30 female patients met the criteria. The mean patient age was 42.16 (range 27–66) years, and the mean patient parity was 1.86 (range 0–5). A total of 14 of these patients underwent fluoroscopic venography. 

Left ovarian vein reflux is more common than on the right, presents earlier than on the right, is easier to evaluate in a shorter period of time, and, thus, was the focus of our study. Anatomical differences largely account for this, including the larger diameter of the left ovarian vein compared to the right and the direct drainage of the right into the inferior vena cava [10]. Right venous reflux is considered a potential consequence of left ovarian vein trunk overload and a possible left-to-right vascular shunt mechanism [11]. By focusing our study on the left ovarian vein, we were able to capture pelvic venous reflux most efficiently.

### 2.2. MRI Acquisition

Patients were evaluated with MRI for ovarian vein reflux with multiphasic dynamic contrast-enhanced MRI (gadobutrol at a rate of 2 mL/s) using DISCO–MRI at 3T (GE, Waukesha, WI, USA) using a 32-channel body array receiving coil. The DISCO–MRI protocol (Table 1) emphasized temporal resolution and spatial resolution with the following protocol: 18 phases, repetition time/echo time= 3.6/1.67 ms, field of view of 38 × 25 cm, slice thickness of 2.8 mm, 226 slices, flip angle of 12°, acceleration of 1.8 phase × 2.5 slice for a total effective acceleration of 4.5, matrix 192 ×160, and BW = ±167 kHz. The time per phase was 3.2 s (range from 3.11 to 3.58 s), leading to a total DISCO sequence time of 63 s (range from 61 to 70 s). In addition, conventional imaging was performed, including coronal and axial T2 weighted single-shot fast-spin echo (SSFSE) imaging, sagittal and oblique axial T2 weighted FSE, and post-contrast in-phase and out-phase imaging.

In the evaluation of ovarian vein reflux, the radiologists with 5 years of post-fellowship experience (blinded to fluoroscopy results) determined if the patient had left ovarian vein reflux by evaluating the post-processed DISCO images using maximum intensity projection (MIP) algorithms in a coronal reformat and determined whether the left ovarian vein filled retrogradely before the filling of the common iliac vein [11]. Each study was evaluated for (1) the presence or absence of ovarian vein dilation or reflux, and if reflux was present, (2) the phase in which the reflux was observed. Cohen’s Kappa analysis measured the level of agreement between DISCO–MRI and fluoroscopic venograms.

## 3. Results

Out of the initial 30 patients who were imaged with DISCO sequences due to concern for pelvic venous syndrome in the clinical record, 16 patients were excluded from analysis (Figure 1). Among the 30 patients, 14 were further evaluated for ovarian vein reflux and underwent the appropriate MRI sequences with DISCO and fluoroscopic venograms. Of these 14 patients, 11 patients had concordant results between the MRI and venogram, and 3 patients had discordant results. In seven of the concordant cases, left ovarian vein reflux was observed prior to filling of the iliac veins with DISCO imaging as well as on the venogram. Among these cases, three showed concurrent left iliac vein compression, and an additional three showed concurrent left renal vein compression. In contrast, four of the concordant cases did not demonstrate left ovarian vein reflux on either DISCO imaging or the venogram. In two of these four cases, left renal compression was noted. The three discordant cases all demonstrated ovarian vein reflux on the venogram but not DISCO imaging. Among these cases, two showed left retroaortic renal compression, and one exhibited left iliac vein compression. Imaging from representative patients is shown in Figure 2, Figure 3 and Figure 4. Cohen’s Kappa demonstrated moderate agreement between DISCO–MRI and the venogram, with κ = 0.57 (95% CI = 0.18, 0.96). Of the seven patients with ovarian reflux detected with DISCO imaging, the average phase of initial reflux detection was phase 7 (range: 5 to 9 phases, 21 to 34 s).

## 4. Discussion 

Pelvic venous disorders are often overlooked as a cause of chronic pelvic pain in women, even though these disorders may contribute to approximately 30% of chronic pelvic pain cases [12,13]. Accurate detection of pelvic vein insufficiency can sometimes be challenging due to anatomical variations and nonspecific symptoms. Imaging is helpful in assessing chronic pelvic pain originating from a pelvic venous disorder. Fluoroscopic venography is considered the most reliable method for diagnosing pelvic venous disorders due to its evaluation of venous hemodynamics during provocative maneuvers such as Valsalva and table tilting. Because of its invasive nature, the procedural resources required, and the potential for exposure to ionizing radiation, it is primarily recommended for patients with strong indications of pelvic venous disorders following non-invasive imaging [5,12]. Reliable, non-invasive imaging methods are needed for the preliminary evaluation of pelvic venous disorders and treatment planning. Transabdominal or transvaginal duplex ultrasound is commonly used for the first-line evaluation of pelvic venous disorders due to its widespread availability and affordability. However, the effectiveness of these techniques can vary based on the technologist’s skill and the patient’s body type. On the other hand, single-phase contrast-enhanced MRI and CT scans are suboptimal for assessing retrograde flow in the ovarian or internal iliac veins, as a continuous dynamic component is helpful in the diagnosis.

Time-resolved MR angiography is a dynamic examination modality that can be used for evaluating pelvic veins and surgical planning by providing information about the direction of venous flow. Dilated ovarian and pelvic veins are best visualized during the venous phase on contrast-enhanced T1-weighted images [14]. Traditionally, these techniques are four-phase techniques with an arterial, late arterial, venous, and late venous. In a study by Yang et al., time-resolved four-phase MR angiography compared to fluoroscopic venography showed specificity, sensitivity, and accuracy of 67%, 100%, and 79% vs. 75%, 100%, and 84%, respectively [8]. According to Attia et al., the sensitivity, specificity, and accuracy of a four-phase TR-MRV for detecting ovarian vein reflux were 87%, 80%, and 84%, respectively, while for internal iliac vein reflux, these values were 75%, 53%, and 72%. For pelvic venous plexus reflux, they were 92%, 69%, and 64% [15]. Traditional TR-MRA/MRV depends on only four phases. Catching a dynamic diagnosis like reflux during one of these specific four phases can be limiting. This is where applying a technique like DISCO–MRI can be helpful. DISCO–MRI is able to quickly sample multiple phases through arterial and venous acquisition (16 phases in this study), with shorter intervals between phases. This allows the exam to more likely catch the optimal phase where reflux may be visualized. Additionally, relatively high-resolution anatomic imaging is collected, which helps to isolate the vessels of interest.

This is a proof-of-concept study of the utility of DISCO–MRI in evaluating pelvic venous disorders. While this study is limited to a small cohort of patients, our data suggest that DISCO–MRI may be a valuable non-invasive tool to assist in diagnosing this condition. This study would benefit from a prospective study with both DISCO–MRI and fluoroscopic venography in a larger group of patients with chronic pelvic pain. One limitation for several patients was the inability to visualize either reflux or normal drainage of ovarian veins. The average scan time was 63 s. Extending the scan time to 90 s may be helpful in capturing variations in patients’ drainage times, as there is some variation in our patient population. Additionally, MRI is performed supine, while fluoroscopic venography can be performed with a slight table tilt simulating a semi-upright position, which is more concordant with physiologic conditions to elicit ovarian vein reflux [16]. Imaging modalities that allow accurate evaluation of anatomical variations and blood flow dynamics in the ovarian veins help to achieve successful treatment. DISCO–MRI can be used as a non-invasive initial workup for pelvic pain before fluoroscopic venography. Future studies will include evaluations with longer dynamic scan times as well as patients with diverse pelvic venous disorders. DISCO–MRI may help with better planning for the treatment of ovarian vein reflux rather than relying on fluoroscopy alone.

## 5. Conclusions

The current study demonstrates that time-resolved DISCO–MRI has the potential to be a non-invasive method for evaluating ovarian vein reflux. The current gold standard is fluoroscopy, and having a reliable technique, such as DISCO–MRI, may help plan for procedures and decrease unnecessary diagnostic fluoroscopic evaluations.

## Figures and Tables

**Figure 1 diagnostics-14-01737-f001:**
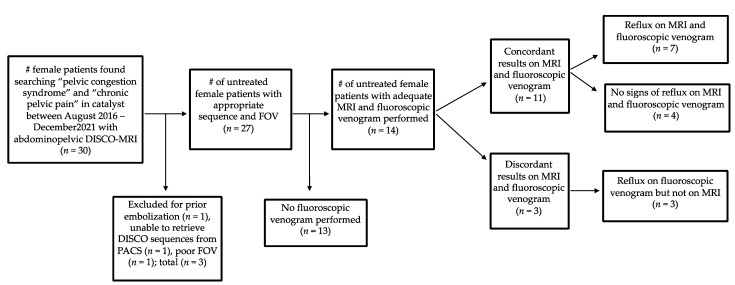
Flow diagram of patients included in the study. DISCO (Differential Subsampling with Cartesian Ordering). # number.

**Figure 2 diagnostics-14-01737-f002:**
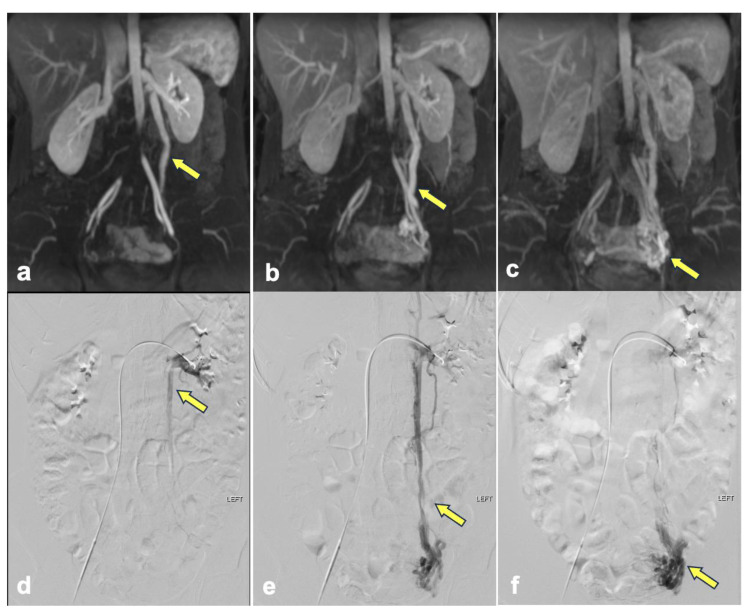
The panels show representative images from a 36-year-old female patient with left ovarian vein reflux. DISCO–MRI phase 7 (**a**), phase 12 (**b**), and phase 18 (**c**) show retrograde flow of contrast medium from the left renal vein to the left ovarian vein (arrows). Left renal fluoroscopic venography (**d**–**f**) shows the dilated left ovarian vein and left para-uterine veins (arrows).

**Figure 3 diagnostics-14-01737-f003:**
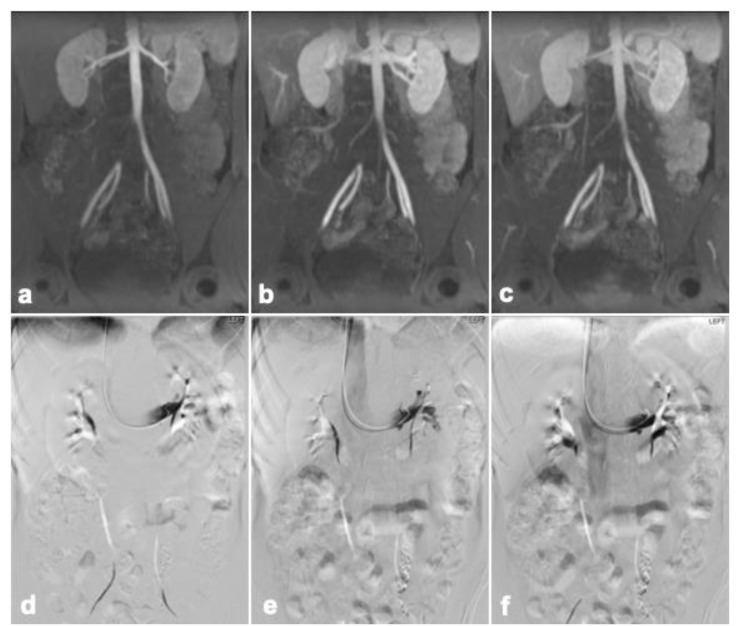
A 33-year-old female patient with no signs of reflux on both DISCO–MRI (**a**–**c**) and fluoro-scopic venography (**d**–**f**). (**a**) DISCO–MRI phase 7, (**b**) phase 12, and (**c**) phase 18.

**Figure 4 diagnostics-14-01737-f004:**
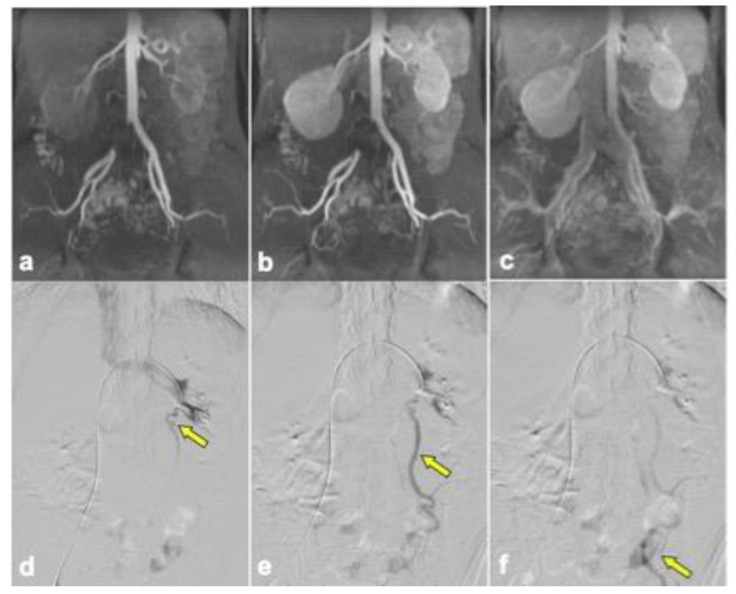
DISCO–MRI of a 43-year-old female with left ovarian vein reflux did not reveal any evidence of left ovarian vein reflux during different phases (**a**–**c**), in contrast to retrograde flow seen in fluoroscopic venography (arrows in **d**–**f**). Drainage up the left ovarian vein on the MRI is not observed, but venous filling in the pelvis without reflux is seen by this expected time point. (**a**) Phase 6, (**b**) phase 8, (**c**) and phase 18.

**Table 1 diagnostics-14-01737-t001:** Protocol guidance for DISCO sequence for pelvic venous disorders.

Parameter	
Field Strength	3.0 T
Total number of phases	18
Echo time (TE)	1.67 ms
Repetition time (TR)	3.61 ms
Field of view	38 × 25 cm
Flip Angle	12°
Number of slices	226
Slice Thickness	2.8 mm
Temporal Resolution	3.2 s
Acceleration	1.8 × 2.5 phase × slice
Matrix	192 × 160
Bandwidth	±167 kHz

## Data Availability

All data relevant to this study are contained within the manuscript. No additional data beyond those presented in the manuscript can be provided.

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
