# Peer review of "Application of Differential Subsampling with Cartesian Ordering in Evaluating Left Ovarian Venous Reflux for Pretreatment Planning for Pelvic Venous Disorders"

_diagnostics, 2024, doi:10.3390/diagnostics14161737_

Round 1

Reviewer 1 Report

Comments and Suggestions for Authors

This is an interesting study. However, it should be clear in the manuscript why there are no cases related to the right ovarian vein (although the reflux in the left ovarian vein is more common than in the right one, as the latter drains directly into the inferior vena cava).

Author Response

1. Summary

Thank you very much for taking the time to review this manuscript. Please find the detailed

responses below and the corresponding revisions/corrections highlighted/in track changes in

the re-submitted files.

2. Point-by-point response to Comments and Suggestions for Authors

Comments 1: This is an interesting study. However, it should be clear in the manuscript why

there are no cases related to the right ovarian vein (although the reflux in the left ovarian vein

is more common than in the right one, as the latter drains directly into the inferior vena cava).

Response 1: Thank you for pointing this out. We agree with this comment. Therefore, we

have added a paragraph about why we focus on left ovarian vein reflux – page 2, paragraph

5, and lines 79-86.

Reviewer 2 Report

Comments and Suggestions for Authors

In this study, authors studied concordance between DISCO-MRI and fluoroscopic venogram, in determining left ovarian venous reflux, and the concordance found was 78.6%. Interesting paper. Beautiful explanations. Some details need to be resolved:

-You did not explain why you chose to study only the left ovarian veins, not the right, too, not both. Please explain.

-In the beautiful flow diagram, you nicely showed how you came to a final number of 14 patients who were explored by both methods. Ok. But from all the patients that you have explored, no one had bilateral or right ovarian vein reflux? It is hard to believe.

-in Materials and methods you wrote: “A retrospective evaluation was performed on studies from August 2016 to December 2021 with the following criteria: abdominopelvic magnetic resonance imaging in a female patient with a reason for scan or study notes including concern for chronic pelvic pain and pelvic congestion syndrome……   30 total female patients met the criteria. The mean patient age was 42.16 (range 27-66) years, and the mean patient parity was 1.86 (range 0-5).14 of these patients proceeded to fluoroscopic venography.” There is hard to believe that form August 2016 to December 2021 you found no patient with right or bilateral ovarian reflux. Please check again and/or write somewhere in the flow diagram that you eliminated x patients with right/bilateral ovarian vein reflux, or please write in the inclusion criteria that you only chose patients with left ovarian vein reflux because….

-in MRI Acquisition, you wrote: “In the evaluation of ovarian vein reflux, the radiologists with 5 years of post-fellow ship experience (blinded to fluoroscopy results) determined if the patient had left ovarian vein reflux by evaluating the post-processed DISCO images using maximum intensity projection (MIP) algorithms in a coronal reformat and determined whether the left ovarian vein filled retrograde before the filling of the common iliac vein” .Ok. Why only the left one? Please explain.

-in the Discussion, as well as in Introduction, you explain about ovarian venous reflux, not only left one. Please explain why you chose only the left one.

-In References, half of them (7 out of 15) were written before 2019. Please replace some of them with some newer ones.   

Author Response

1. Summary

Thank you very much for taking the time to review this manuscript. Please find the detailed

responses below and the corresponding revisions/corrections highlighted/in track changes in

the re-submitted files.

2. Point-by-point response to Comments and Suggestions for Authors

Comments 1: You did not explain why you chose to study only the left ovarian veins, not the

right, too, not both. Please explain.

Response 1: Thank you for pointing this out. We agree with this comment. Therefore, we

have added a paragraph about why we focus on left ovarian vein reflux – page 2, paragraph

5, and lines 79-86.

Comments 2: In the beautiful flow diagram, you nicely showed how you came to a final

number of 14 patients who were explored by both methods. Ok. But from all the patients

that you have explored, no one had bilateral or right ovarian vein reflux? It is hard to

believe.

Response 2: Evaluating bilateral or right ovarian vein reflux was beyond the scope of our

study. We have added a paragraph explaining why we focused on left ovarian vein reflux –

see page 2, paragraph 5, lines 79-86

Comments 3: in Materials and methods you wrote: “A retrospective evaluation was

performed on studies from August 2016 to December 2021 with the following criteria:

abdominopelvic magnetic resonance imaging in a female patient with a reason for scan or

study notes including concern for chronic pelvic pain and pelvic congestion syndrome……

30 total female patients met the criteria. The mean patient age was 42.16 (range 27-66) years,

and the mean patient parity was 1.86 (range 0-5).14 of these patients proceeded to

fluoroscopic venography.” There is hard to believe that form August 2016 to December 2021

you found no patient with right or bilateral ovarian reflux. Please check again and/or write

somewhere in the flow diagram that you eliminated x patients with right/bilateral ovarian

vein reflux, or please write in the inclusion criteria that you only chose patients with left

ovarian vein reflux because….

Response 3: We have added a paragraph about why we focus on left ovarian vein reflux –

page 2, paragraph 5, and lines 79-86.

Comments 4: In MRI Acquisition, you wrote: “In the evaluation of ovarian vein reflux, the

radiologists with 5 years of post-fellow ship experience (blinded to fluoroscopy results)

determined if the patient had left ovarian vein reflux by evaluating the post-processed

DISCO images using maximum intensity projection (MIP) algorithms in a coronal reformat

and determined whether the left ovarian vein filled retrograde before the filling of the

common iliac vein” .Ok. Why only the left one? Please explain.

Response 4: We have added a paragraph about why we focus on left ovarian vein reflux –

page 2, paragraph 5, and lines 79-86.

Comments 5: In the Discussion, as well as in Introduction, you explain about ovarian venous

reflux, not only left one. Please explain why you chose only the left one.

Response 5: We have added a paragraph about why we focus on left ovarian vein reflux –

page 2, paragraph 5, and lines 79-86.

Comments 6: In References, half of them (7 out of 15) were written before 2019. Please

replace some of them with some newer ones.

Response 6: Thank you for pointing this out. We have added a more recent reference, and

now over 62% of them are from 2019 or later.
